# Effect of Sea Level Rise and Access Channel Deepening on Future Tidal Power Plants in Buenaventura Colombia

**Juan Gabriel Rueda-Bayona** [1,2,*] **, José Luis García Vélez** [1,2] **and Daniel Mateo Parrado-Vallejo** [1,2]

[1]  Natural and Environmental Resources Engineering School (EIDENAR), Faculty of Engineering, Universidad del Valle, Cali 760042, Colombia
[2]  INCIMAR Institute of Marine Sciences and Limnology, Universidad del Valle, Cali 760042, Colombia
*   Correspondence: juan.gabriel.rueda@correounivalle.edu.co or ruedabayona@gmail.com

**Abstract:** The evolution of tidal stream turbines is increasing the feasibility of future tidal plants in shallow depth areas with mid-tidal ranges (<5 m). However, extreme events such as changes in bathymetry due to the access channel deepening of coastal ports and sea level rise modify hydrodynamics and might affect the infrastructure and energy production of tidal energy converters. This research focused on Buenaventura Bay to analyze the effect of these extreme events on marine currents through calibrated-validated numerical modeling. Several monitored points were analyzed, and the results highlighted that the bay has potential for implementing tidal stream turbines because of the reported velocities between 0.25 and 2 m/s. The sea level rise increased 11.39% and access channel deepening reduced by 17.12% the velocity currents of the bay, respectively. These findings convert Buenaventura Bay to a candidate for implementing third generation tidal stream turbines and motivate future research for implementing tidal power systems in crucial areas such as the Colombian Pacific, where communities face restrictions in accessing affordable and clean energy.

**Keywords:** deepening; hydrodynamics; estuary; marine currents; sea level rise; tidal energy

## 1. Introduction

Decarbonizing electricity generation and the diversification of the energy matrices of economies are two of the most important priorities worldwide [1]; hence, non-conventional renewables such as marine energies are being considered more seriously nowadays [2]. The dependency on electricity generation from traditional gas–carbon sources not only affects the competitivity of economies when international prices of importation rise, but also puts social wellness at risk when energy access is limited or blocked from other countries' suppliers. In this sense, energy transition to a clean matrix will be successful when the new renewable technologies such as offshore wind and marine energies consider the relevance of human health, the natural environment and resources [3].

Marine energies are considered non-conventional since the main source of energy comes from waves, thermohaline gradients and tides. From these marine renewables, tidal technologies can extract the kinetic and potential energy of the sea and are grouped in two main categories: 1- tidal stream turbines and 2- tidal barrages. Tidal stream turbines are independent horizontal or vertical axial turbines that extract energy from the currents generated by tides, and tidal barrages requires a dam or reservoir to provoke hydraulic heads for activating turbines beneath chambers or gates [4].

The first application of tidal technology in the world was a coastal hydroelectric dam inaugurated in 1968 as La Rance tidal plant in France. Later, Russia applied tidal barrages in Kislogubsk or Kislaya Guba (1968), China inaugurated the Jiangxia plant in 1980, Canada activated the Annapolis plant (1981) and South Korea built the Sihwa tidal plant (2011) [4]. Other configurations of tidal technologies derived from tidal barrages and tidal stream turbines are described by Shetty and Priyam [4]. Chowdhury et al. [5] reviewed the current

trends and projects of tidal energy technology and pointed out that the worldwide ocean power generation scenario 2000–2030 (according to the International Energy Agency) is about 15 TWh by 2030. The research also reported international projects in the USA, Norway, the UK, Australia, The Netherlands, France, South Korea, Spain, Sweden, Canada, Ireland and Germany developing tidal energy technologies. Vogel et al. [6] analyzed the prospects for tidal Stream Energy in the UK and South America, and concluded that Brazil, Chile, Uruguay and Argentina have locations with high potential for tidal energy projects.

The report of IRENA [7] showed a world map of average tidal range pointing to a 3–4 m tidal range in the Colombian Pacific, and Khan et al. [8] mentioned that spring tides with a range between 4 and 12 m have the potential to produce 1–10 MW/km of electricity. Recently, Quintero and Rueda-Bayona in 2021 [9] reported a potential of 19,360 Wh/month in the central zone of the Colombian Pacific coast through a tidal stream with 1 m of sweep area in a tidal range of 3–4 m. Also, the evolution of third generation tidal stream turbines is widening the possibilities for producing electricity in shallow water areas with current velocities below 2 m/s [10,11]. In this sense, there exists a potential for generating electricity from tides in Colombia that must be studied with more attention.

The evaluation of tidal energy sources requires the characterization of water levels and currents because they are the main parameters for the power calculations [12,13]. When measured data are limited, the numerical approaches solve the lack of information in tidal power projects through the application of validated-calibrated hydrodynamic models [14], theoretical-parameterized equations [15], or Computational Fluid Dynamics (CFD) modeling for evaluating the hydraulic properties of tidal power plants [16–18]. Two-dimensional hydrodynamic model results have been analyzed to explore the impact of dikes on tidal currents [19,20], and other studies have evaluated tidal Stream Energy resources using 3D numerical modeling [21–24]. Also, to identify potential areas for tidal plants the Geographic Information System (GIS) and Multi-Criteria Decision Making (MCDM) have been applied [25]. All these numerical approaches and methods have evaluated the tidal energy resource not considering long-term effects (sea level rise, channel deepening) over the main hydraulic parameters.

Several studies focused on the design and operation of tidal power plants, considering technical-economic methods for increasing the economic value [25] and the optimization of the generated power through the selection of a proper number of turbines, gates and other components of a tidal power plant [26]. Other research has reported the use of optimization approaches, e.g., gradient-based optimization techniques to determine the optimal control strategy for several tidal cycles [27], dynamic programming algorithms for the least cost of energy generation [28] and the levelized cost of energy assessments of tidal energy via continuous wavelet transform methods [29]

According to Nevermann et al. [30], a sea level rise of 1.04 m for the year 2100 will impact 86 municipalities along the Colombian coast. In addition, Koks et al. [31] reported that global sea level rise will impact existing and future coastal infrastructures, and Sangsefidi et al. [31] pointed out that sea level rise will shift the coastline into the land, provoking cliff collapse and beach erosion that could damage coastal infrastructure. Khojasteh et al. [32] pointed out that the potential of tidal energy extraction within an estuary—whether through tidal stream turbines or barrages—is influenced by modifications of geomorphology and hydrodynamics due to the effects of sea level rise. These changes may reduce or increase the potential depending on the tidal prism, range, asymmetry, among other factors of the estuary. Considering the recent review of Li and Pan [33], it is possible that currently operating tidal barrage plants and others under development, i.e., those located in Zhejiang Province, China, have not taken into account the effect of sea level rise rates on their tidal plant operation.

The reviewed literature does not report information about the effect of sea level rise, nor access channel deepening for optimizing the navigability of maritime ports. Only the study of Chen and Liu [34] explicitly provides information of the effect of sea level rise (SLR) on tidal energy output. That study validated a 2D hydrodynamic model with

measured data through an Acoustic Doppler profiler (ADCP) with datasets < 20 days. That research could be considered limited because the effects of Sizygy and Quadrature over water level and current dynamics were not fully considered because the dataset did not cover a full moon cycle; Sizygy is the maximum tidal height provoked by the alignment of the sun and moon with respect to the Earth, and Quadrature is the lowest tidal height when sun and moon are 90 degrees with respect to the Earth. Despite the limitations of that study, the research provided valuable information for future tidal power plants in the Penghu Islands, Taiwan. The study of Quintero and Rueda-Bayona [9] was an important steep to motivate the research into tidal power in Colombia; however, there exist gaps in the understanding of hydrodynamics when they are affected by the effects of sea level rise and the modification of coastal morphology through human interventions such as access channel deepening.

Bearing in mind the increasing worldwide interest in marine energies, recent reports of tidal potential energy in Colombia, together with the interest of national government in the energy transition through renewable energies, this study aims to increase the knowledge of tidal energy potential in the coastal and estuarine waters of the central zone of the Colombian Pacific. This research utilized measured oceanographic data to calibrate a 3D hydrodynamic model for simulating one year (2021) of the hydrodynamics of Buenaventura Bay. The results provide information about how the sea level rise and access channel deepening affect the hydrodynamics of coastal current velocities, which is valuable information for the planning and operation of future tidal plants not only in Colombia, but also in other areas in the world.

## 2. Materials and Methods

The domain area of this research is bounded by Buenaventura Bay, located in the central zone of the Colombian Pacific (Figure 1). In the bay is located the most important port of Colombia, which manages 60% of the imported-exported commodities of Colombia and is in the top 10 of Latin American ports [35].

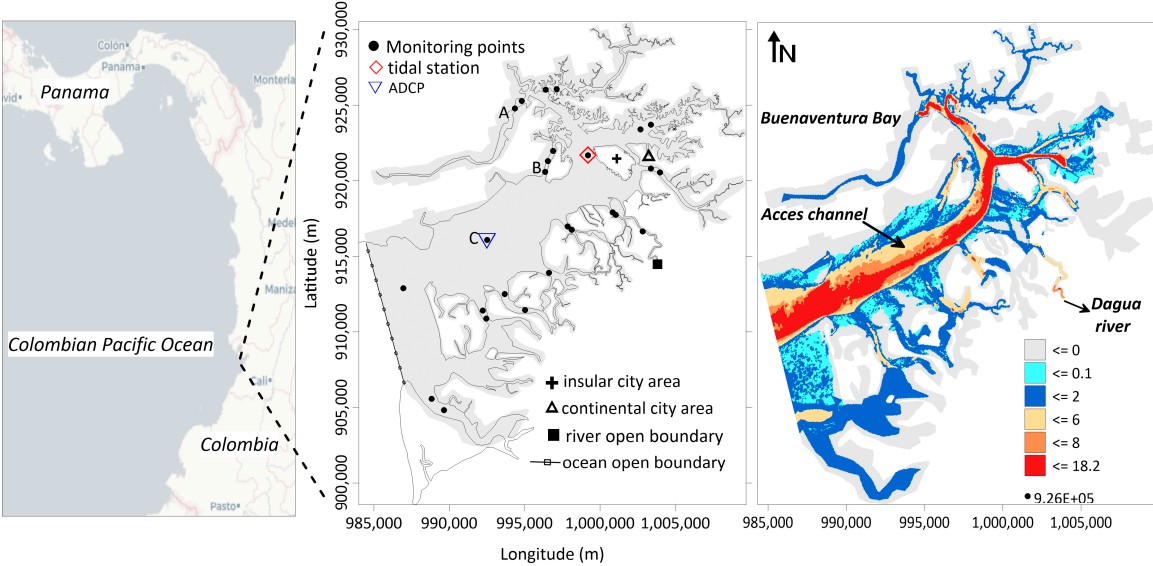

**Figure 1.** Study area, location of monitoring points and domain of the 3D-hydrodynamic model. Coordinates in Magna Sirgas Colombia West unit system. The labels A (Aguadulce creek), B (Crab Island) and C (channel's talweg) denote model monitoring points utilized for further tidal analysis. The Tidal station and ADCP measurements have the coordinates 77.08079 W/3.891239 N and 77.146 W/3.837 N, respectively.

*Model Setup and Data Inputs*

This study implemented the Delft3D model, which is a 3D hydrodynamic model developed by Deltares [36] and has been utilized successfully in Colombia and worldwide [37–39]. The model has 10 layers with sigma vertical coordinates, with a regular mesh grid of 40 m of cell size; sigma vertical coordinates follow the bathymetry no matter the water depth with the same number of vertical points in the grid domain. Figure 1 shows the open boundaries, the locations of the water level station and the ADCP deployment for calibration-validation, and 24 virtual monitoring points of the numerical model; in Table 1 are listed the main parameters of the implemented model. The Grenoble model [40] was used to provide simulated water level data for the ocean open boundary of the Delft3D model; the type of the open boundary condition was "water-level" with the forcing type "time-series".

**Table 1.** Tuned parameters of Delft3D model.

| Parameter | Method/Value |
|---|---|
| Bottom roughness | Chezy/65 |
| Horizontal eddy viscosity | $1 \text{ m}^2/\text{s}$ |
| Horizontal eddy diffusivity | $10 \text{ m}^2/\text{s}$ |
| Turbulence | 3D/k-epsilon |
| Time step | 0.5 min |
| Water density | $1023 \text{ kg/m}^3$ |
| Air density | $1 \text{ kg/m}^3$ |
| Depth at grid cell faces | Mean |
| Threshold depth | 0.1 m |
| Marginal depth | $-7$ m |
| Smoothing time | 60 min |
| Advection scheme for momentum-transport | Cyclic |
| Wind drag coefficients | Breakpoint A/0.000063, 0 m/s<br>Breakpoint B/0.000723, 10 m/s<br>Breakpoint C/0.000723, 20 m/s |

We used the WOA2018 data base (https://odv.awi.de/data/ocean/ (accessed on 1 December 2022) and the official water quality report of 2021 in Buenaventura Bay released by INVEMAR [41] to retrieve information of the thermohaline properties of the study area. Measured in situ data for the calibration and validation of the model of water level (hourly interval), ADCP currents (5 min interval), surface winds (hourly interval) and bathymetry were provided by the General Maritime Directorate of Colombia [42].

The study of Consorcio Dragado Buenaventura [43] reported information to enhance the port capacity for receiving ships with high draught, through the deepening of the access channel with periodic dredging at about 20.5 m of depth. The sea level rise has been estimated through different methods around the world [44], where the rate of rise differs from very low < 0 mm/year to very high > 9 mm/year. However, the research of Gallego-Perez and Selvaraj [44] set a rise of 2.2 mm/year arguing that this rate agreed with other similar studies performed in Buenaventura Bay; hence, we selected this rate for a horizon of 200 years, which represents an increment of 0.44 m in the mean water level of the study area; the 0.44 m was added to the water depths of the hydrodynamic model to represent the future sea level rise.

Consequently, this study stablished 5 hydrodynamic modeling cases as follows:

1. Natural conditions of the bay during 2021.
2. Extra deepening of 5 m in the access channel.
3. Sea level rise of 0.44 m after 200 years with 2.2 mm/year rate.
4. Sea level rise of 0.22 m after 100 years with 2.2 mm/year rate.
5. Sea level rise of 0.11 m after 50 years with 2.2 mm/year rate.

### 3. Results

The results of the model calibration-validation process are presented below, with the harmonic analysis of the water level data measured in the tidal station. Next, we analyzed the numerical results of current velocity of the 24 monitoring points of the study area (Figure 1) to select representative locations with tidal energy potential for the further analysis of the sea level rise and deepening effect.

#### 3.1. Tidal Harmonics Analysis

Ten years of water level measured data with an hourly time interval measured within Buenaventura Bay (Figure 1) were analyzed through Fourier Analysis performed in the software T_TIDE [45]. Constituents with Signal to Noise Ratio (SNR) > 10 were discarded according to the recommendations of Pawlowicz et al. [45]. Several tidal constituents with semi-diurnal, diurnal and higher order periods were identified as seen in Table 2. The main identified constituents were similar to those reported by Otero, L. [46], who analyzed measured hourly water level data over 47 years (1953–2000). The form factor of tides was calculated as $F = \frac{C\_K_1 + C\_O_1}{C\_M_2 + C\_S_2}$ to identify the tidal regime and it was found that the study area has a F = 0.066743, which represents a semidiurnal tidal regime, results that agreed with other local studies [47].

**Table 2.** Main constituent tidal harmonics of Buenaventura Bay calculated from measured water level data between 2011 and 2021.

| Harmonic | Period (h) | Frequency (cph) | Amplitude (m) | Phase (°) | SNR |
|---|---|---|---|---|---|
| SA | 8766.23148 | 0.000114074 | 0.120153074 | 234.8465377 | 54.82042061 |
| SSA | 4382.9063 | 0.000228159 | 0.070128659 | 139.2832099 | 20.75563454 |
| O1 | 25.81934166 | 0.038730654 | 0.022875503 | 2.017207422 | 65.44156092 |
| P1 | 24.06589023 | 0.041552587 | 0.035745417 | 342.142001 | 119.3355136 |
| S1 | 23.99999686 | 0.041666672 | 0.016676698 | 260.1614375 | 22.69151675 |
| K1 | 23.93446959 | 0.041780746 | 0.104467627 | 347.8688574 | 1534.331354 |
| J1 | 23.0984767 | 0.043292898 | 0.01004813 | 21.27610552 | 10.33498298 |
| EPS2 | 13.12726743 | 0.076177316 | 0.015830933 | 93.69213311 | 14.98524996 |
| 2N2 | 12.90537447 | 0.077487097 | 0.03795249 | 60.73369937 | 81.64286494 |
| MU2 | 12.8717576 | 0.077689468 | 0.049905025 | 94.47466275 | 148.6021144 |
| N2 | 12.65834823 | 0.078999249 | 0.316093425 | 83.74429212 | 5371.587782 |
| NU2 | 12.62600437 | 0.07920162 | 0.057155105 | 87.07549121 | 237.119084 |
| M2 | 12.4206012 | 0.080511401 | 1.506794911 | 109.5253561 | 131116.1652 |
| LDA2 | 12.22177416 | 0.081821181 | 0.017621982 | 86.4619221 | 13.45786245 |
| L2 | 12.19162018 | 0.082023553 | 0.041684396 | 114.8169503 | 80.83575438 |
| T2 | 12.01644919 | 0.083219259 | 0.029581004 | 160.8067384 | 49.3869777 |
| S2 | 12 | 0.083333333 | 0.401177467 | 163.2452598 | 9879.122904 |
| K2 | 11.9672348 | 0.083561492 | 0.089005163 | 161.782011 | 474.3902567 |
| MO3 | 8.386302962 | 0.119242055 | 0.004550463 | 25.77655677 | 11.73695878 |
| M3 | 8.280400802 | 0.120767101 | 0.006672359 | 171.4722701 | 20.37783732 |
| SK3 | 7.992705566 | 0.12511408 | 0.010186052 | 298.6079344 | 52.31519474 |
| MN4 | 6.269173901 | 0.159510649 | 0.027602629 | 267.5590126 | 252.6433172 |
| M4 | 6.210300601 | 0.161022801 | 0.06734016 | 286.8913267 | 1359.285599 |
| SN4 | 6.160192781 | 0.162332582 | 0.011716516 | 2.774502908 | 51.27938999 |
| MS4 | 6.103339275 | 0.163844734 | 0.04394011 | 351.7890852 | 819.7642463 |
| MK4 | 6.094851995 | 0.164072893 | 0.010045556 | 347.8552665 | 32.82585007 |
| S4 | 6 | 0.166666667 | 0.008064286 | 66.99382076 | 24.90401384 |
| 2MK5 | 4.930880214 | 0.202803548 | 0.002693228 | 45.23875797 | 11.18939991 |
| 2MS6 | 4.092387536 | 0.244356135 | 0.00661569 | 336.4132609 | 42.50528221 |
| 2SM6 | 4.045666393 | 0.247178067 | 0.004248617 | 13.99209813 | 17.06704416 |
| M8 | 3.105150301 | 0.322045603 | 0.009453122 | 260.3789437 | 207.6626631 |

*3.2. Model Calibration*

The first numerical results showed that modeled water level data at the tidal station (Figure 1) evidenced a difference of about 1 m in the amplitude of high and low tides in the days of Sizygy and 0.2 m in Quadrature, respectively. These differences have been identified previously in the research of Quintero and Rueda-Bayona [9] and Otero, L. [45], who warned of these issues in their numerical modeling. Buenaventura Bay is considered an estuary [48,49], and the tidal kinetic energy is dissipated by the bottom friction and the reduction of the surface control area due to its convergence in shape. Herein, the estuary may be classified as hyper-synchronous because the convergence effects are greater than the frictional effects that provoke a tidal amplification [50,51]. According to this, the amplitude differences found in this study and in the literature showed that hydrodynamic models may be limited to simulate the tidal amplification; hence, here we present an alternative to adjust the input tidal harmonics to solve this issue (Table 3).

**Table 3.** Harmonics for adding tidal amplification in hydrodynamic modeling.

| Harmonic | Period (h) | Amplitude (m) | Phase (°) |
|----------|-----------|---------------|-----------|
| 1 | 12 | 0.015 | 0.785531 |
| 2 | 24 | 0.015 | 0.785531 |
| 3 | 12.42 | 0.25 | 0.785531 |

The mentioned differences of amplitude during modeling may occur when the input tidal parameters for the hydrodynamic model come from offshore points (outside the bay) and the control point of measured data is in the bay at the upper reaches. Then, for correcting the differences in water level modeling we added three additional harmonics (Table 3) to the ocean open boundary of the model. These three harmonics were found by adding several harmonics to the input water level data, until the modeled and measured water levels agreed and the amplitude errors reduced. We took the harmonic Mf reported by Otero, L. [45] as a reference because the initial errors in amplitude were higher every [46] 328 h (13.55 days), related to the changes in Sizygy and Quadrature.

After the addition of the three harmonics, the modeled water level amplitudes were corrected, and the results are depicted in Figure 2. The statistical results of calibration showed good results with a high correlation coefficient and a significative *p*-value and low RMSE (Figure 2a), and the behavior of the modeled water level during the year showed synchrony and coherence with the measured water level (Figure 2b,c). The amplitudes were fine in phase with differences less than 0.1 m in scarce events of high and low tides; what denotes tidal amplification was considered by the model.

To validate the capability of the model in simulating the hydrodynamics, we compared the numerical results of current velocity at the surface layer against the velocities measured by the ADCP (Figure 1) located in a depth zone of 10 m according to the measured bathymetry data. The comparison of the horizontal velocity of Figure 3a,b showed a good fit in magnitude (R = 0.65, *p*-value < 0.05) and a correlation of R = 0.80 in direction with *p*-value < 0.05. As a result, the behavior of currents was associated to the tidal regime, and the modeled results not only fit in magnitude, but also were good in amplitude and periodicity of direction (Figure 3c).

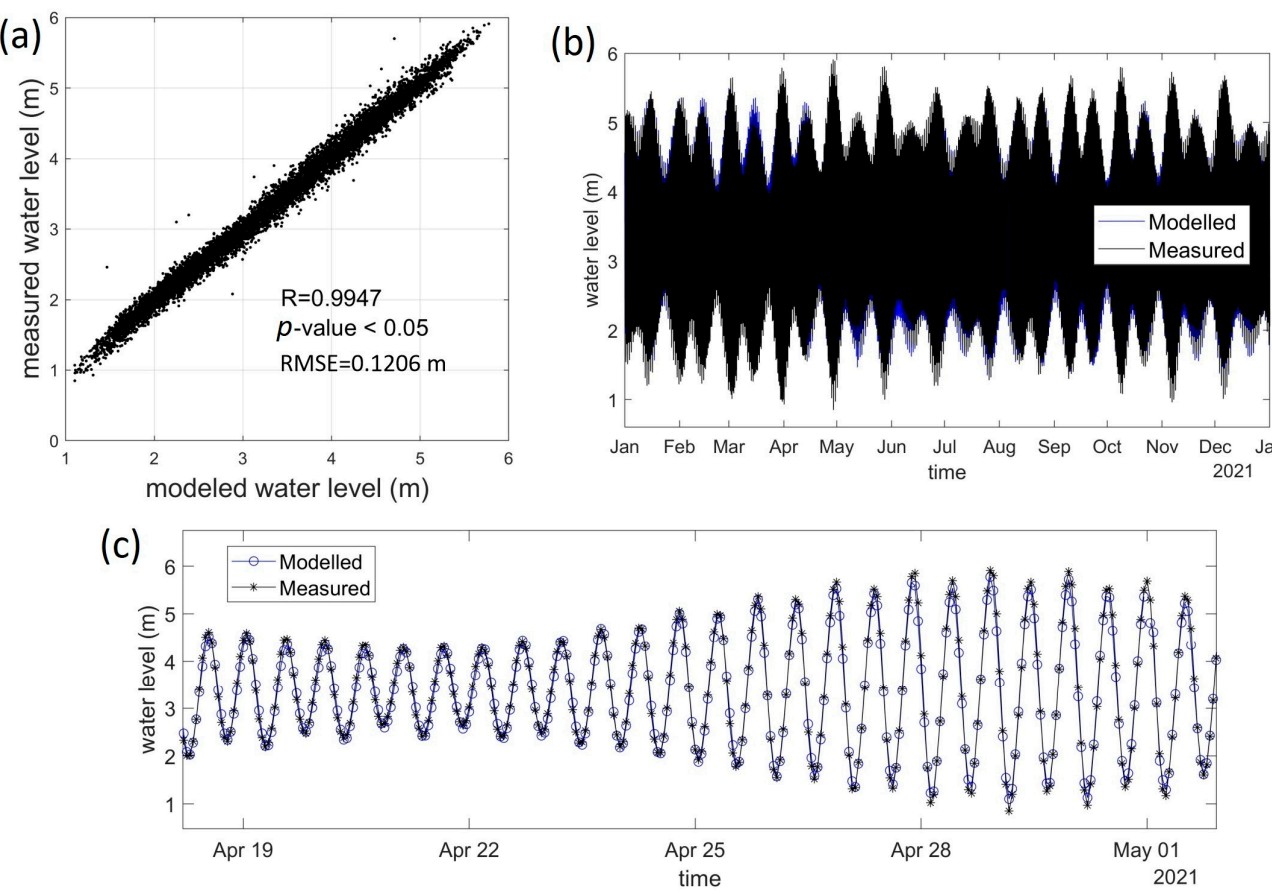

**Figure 2.** Model calibration results of water level data: (**a**) diagram dispersion modeled vs. measured, (**b**) hourly 1-year (2021) time series comparison, (**c**) zoom-in of time series comparison.

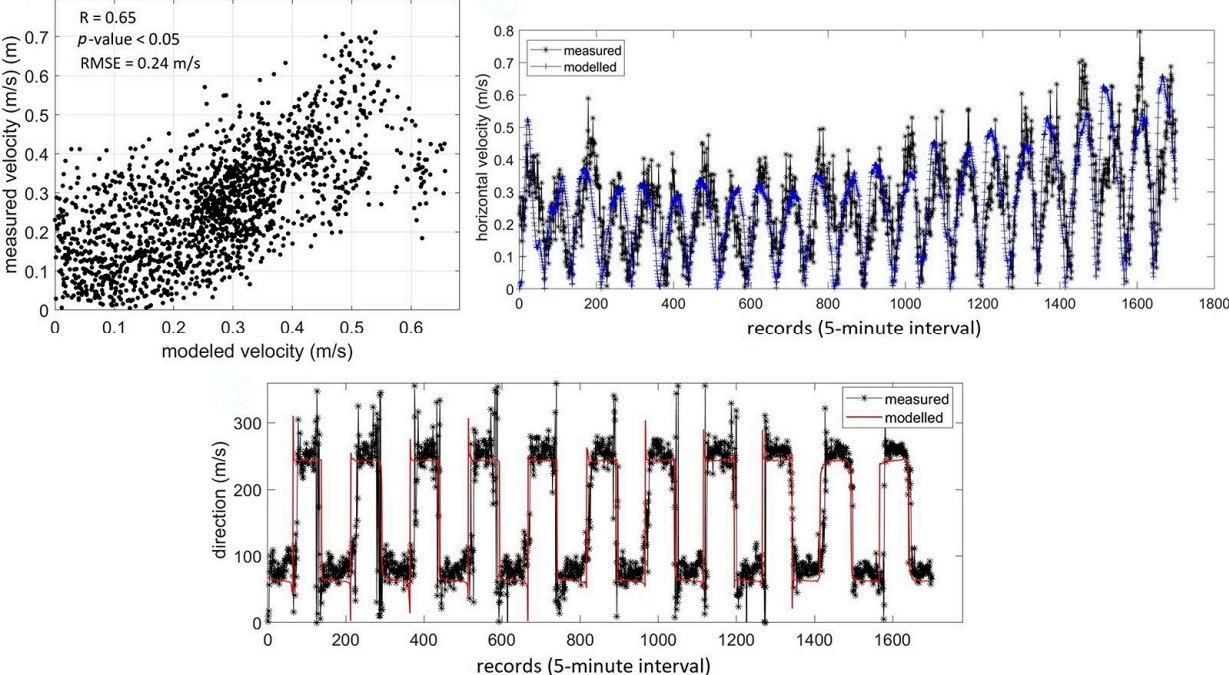

**Figure 3.** Comparison of modeled-measured horizontal velocity at first sigma layer (1 m from surface) with 5 min time interval for the period 19 April 2021, 15:00:00 h to 25 April 2021–12:25:00 h.

### 3.3. Effect of Sea Level Rise and Access Channel Deepening

The hydrodynamic field in Buenaventura Bay is governed by tidal cycles; hence, we select the highest Sizygy and lowest Quadrature of 2021 to analyze the response of currents during the highest and lowest water level events. In Figure 4 are depicted the currents during ebb tides, where the maximum velocities were in the coastal channels (estuaries) with maximum velocities up to 2.2 m/s. The access channel reported high velocities of about 0.8 m/s and low velocities between 0.1 and 0.25 m/s.

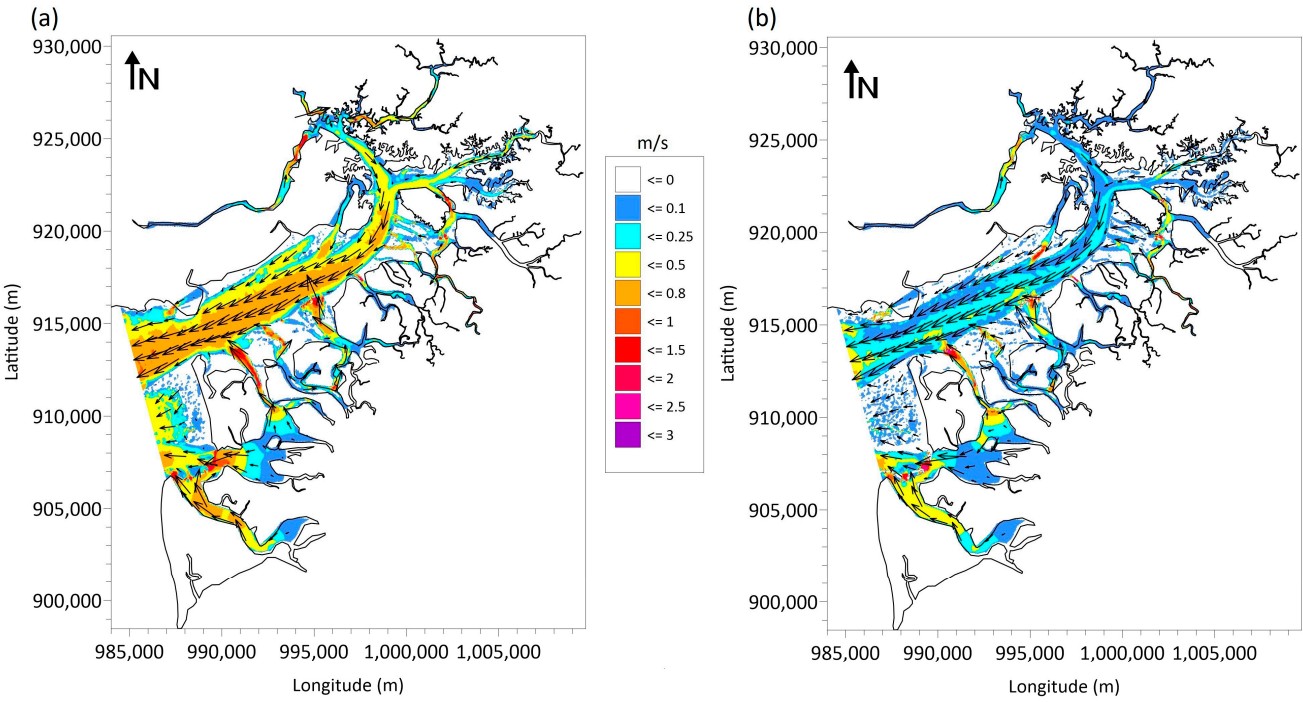

**Figure 4.** Modeled hydrodynamic field during ebb tide at surface layer (1 m below surface) for the: (**a**) highest Sizygy (29abr2021–00:00 h) and (**b**) the lowest Quadrature (22dic2021–14:00 h) in Buenaventura Bay.

The 24 monitoring points were analyzed and shared similar patterns, with three mean velocities (0.25, 0.5 and 1 m/s), fluctuating according to the tidal regime of Buenaventura Bay. As a result, we selected three representative monitoring points labeled as A, B and C (Figure 1) to inspect their behavior during the most important Sizygy and Quadrature events of 2021. The results of the five modeling cases for normal, sea level rise (SLR) and deepening conditions are summarized in Table 4. The modeling case with SLR of 200 years was the most important compared to SLRs of 50 and 100 years, respectively, which provoked a positive effect over the currents, and the deepening case showed a negative effect, respectively. In line with the above, the modeled currents of lowest Quadrature are plotted in Figure 5 for the monitoring points A, B and C (Figure 1). Point A (Figure 5a) in normal conditions showed maximum velocities with a mean value of 1 m/s, maximum up to 1.7 m/s during spring-ebb tides and low values around 0.5 m/s during high-low tides. Points B and C (Figure 5b,c) showed mean values of current velocity of 0.25 and 0.5 m/s, respectively; point C, which is located within the main access channel, clearly followed the semidiurnal tidal regime. It was observed in the results of all three points that sea level rise increased the mean velocities, and the channel deepening reduced them.

**Table 4.** Statistics of modeled velocity currents for the highest Syzygy and lowest Quadrature of 2021 in Buenaventura Bay. Red-blue colors represent decrement-increment of velocity with respect to normal conditions. The label % variation represents de increment or decrement of velocity currents with respect to normal conditions.

| | Quadrature (Lowest Tides) | | | | | | | | | | | | | | | | | | |
|---|---|---|---|---|---|---|---|---|---|---|---|---|---|---|---|---|---|---|---|
| | Normal conditions | | | Channel deepening | | | | Sea level rise (200 yr) | | | | Sea level rise (100 yr) | | | | Sea level rise (50 yr) | | | |
| Point | mean | min | max | mean | min | max | % variation | mean | min | max | % variation | mean | min | max | % variation | mean | min | max | % variation |
| A | 0.91 | 0.01 | 1.72 | 0.93 | 0.19 | 1.65 | 1.55 | 1.05 | 0.05 | 1.77 | 14.66 | 1.00 | 0.38 | 1.80 | 9.29 | 0.97 | 0.32 | 1.78 | 6.06 |
| B | 0.15 | 0.01 | 0.42 | 0.12 | 0.02 | 0.32 | −18.96 | 0.17 | 0.01 | 0.49 | 9.85 | 0.17 | 0.03 | 0.50 | 8.97 | 0.16 | 0.01 | 0.46 | 4.59 |
| C | 0.39 | 0.03 | 0.79 | 0.31 | 0.01 | 1.72 | −20.74 | 0.42 | 0.02 | 0.80 | 7.92 | 0.44 | 0.03 | 1.78 | 11.93 | 0.43 | 0.04 | 1.76 | 9.81 |
| | Syzygy (lowest tides) | | | | | | | | | | | | | | | | | | |
| | Normal conditions | | | Channel deepening | | | | Sea level rise (200 yr) | | | | Sea level rise (100 yr) | | | | Sea level rise (50 yr) | | | |
| | mean | min | max | mean | min | max | % variation | mean | min | max | % variation | mean | min | max | % variation | mean | min | max | % variation |
| A | 1.07 | 0.11 | 2.13 | 1.07 | 0.36 | 2.05 | 0.10 | 1.20 | 0.18 | 2.31 | 12.05 | 1.07 | 0.09 | 2.11 | −0.28 | 1.03 | 0.09 | 2.12 | −3.83 |
| B | 0.20 | 0.00 | 0.53 | 0.15 | 0.00 | 0.52 | −22.77 | 0.23 | 0.00 | 0.54 | 17.13 | 0.22 | 0.01 | 0.50 | 8.25 | 0.21 | 0.01 | 0.48 | 3.39 |
| C | 0.57 | 0.01 | 1.16 | 0.41 | 0.04 | 0.97 | −28.69 | 0.60 | 0.01 | 1.27 | 5.01 | 0.57 | 0.03 | 1.05 | −0.48 | 0.56 | 0.04 | 1.04 | −1.84 |

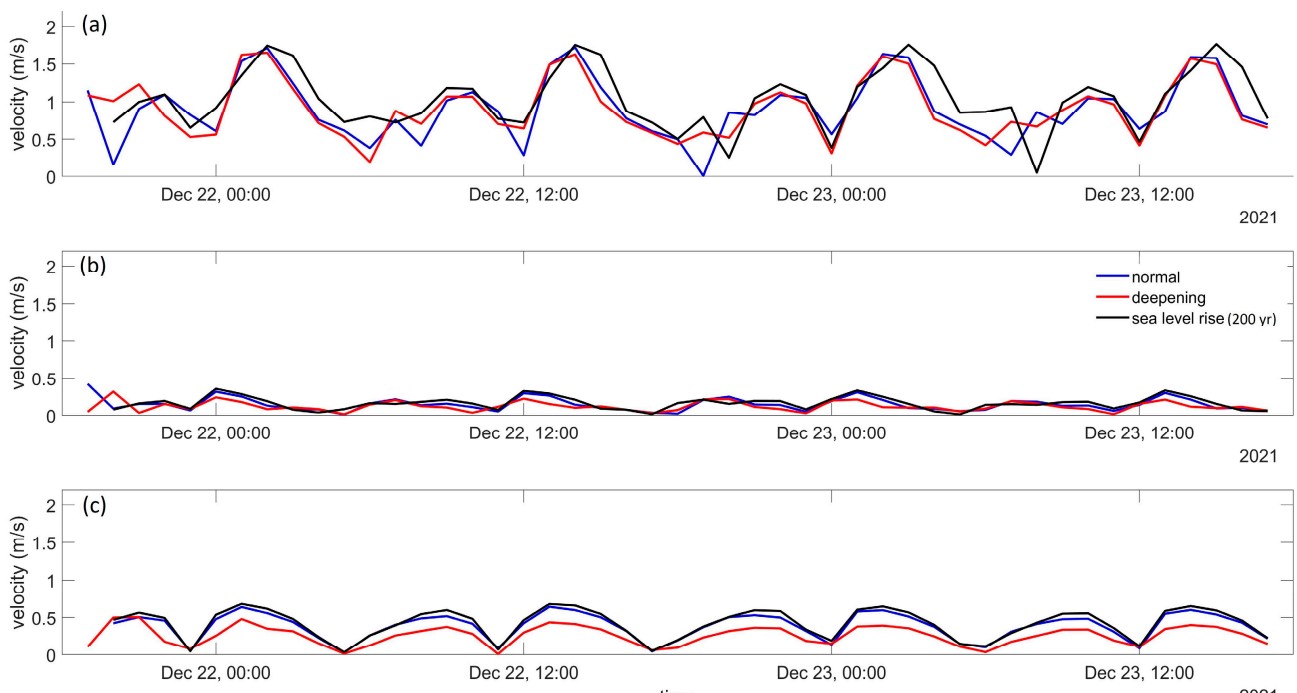

**Figure 5.** Modeled velocity currents at surface layer for the lowest Quadrature of 2021 in Buenaventura Bay of the monitoring points (**a**) A, (**b**) B and (**c**) C.

For the highest Sizygy of 2021, the numerical results of normal conditions pointed out that the highest mean velocities occurred in point A (2.1 m/s), followed by point C (0.75 m/s) and point B (0.4 m/s) during the spring-ebb tides (Figure 6). The results of normal conditions during the highest Sizygy (Figure 6) evidenced that the velocity currents were higher than the results of the lowest Quadrature (Figure 5), which is an expected behavior because of the higher water level amplitudes of Sizygy. The results of sea level rise and deepening conditions in Sizygy (Figure 6) showed higher mean velocities compared to the results of the same conditions in Quadrature (Figure 5), evidencing that the higher water levels of Sizygy increased the velocity currents for all the three modeling cases (normal, sea level rise and deepening). According to the results of Figure 6, the sea level rise and access channel deepening increased and reduced by about 14.28% the velocity currents, respectively.

In Table 4 are listed the statistical results of the three monitoring points retrieved from the five numerical cases: normal condition, channel deepening and sea level rise (50, 100 and 200 years). The results evidenced that points A and B reported the highest and lowest mean velocities during the most important Quadrature and Syzygy events of 2021. To ease the identification of the effects of the channel deepening and sea level rise on the currents we calculated the percentage variation of the velocity currents with respect to normal conditions. As a result, the mean behavior of the monitoring points in Quadrature and Sizygy evidenced that channel deepening reduced the mean velocities in point B and C, but not point A, with a low increment of 1.55% and 0.10%.

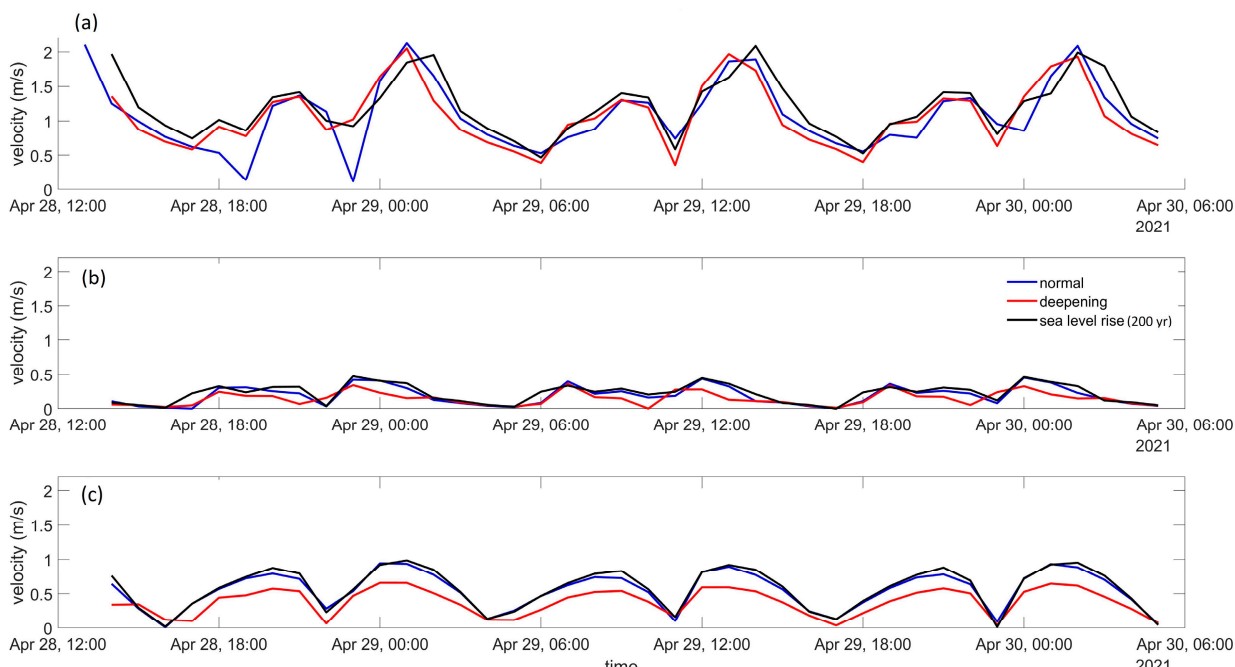

**Figure 6.** Modeled velocity currents at surface layer for the highest Sizygy of 2021 in Buenaventura Bay of the monitoring points (**a**) A, (**b**) B and (**c**) C.

In the case of SLR modeling cases, all the monitoring points reported positive variations (5.01–17.13%) of the velocity currents during Quadrature and Sizygy for 200 years (Table 4). The results for 50 and 100 years showed positive variations (4.59–11.93%) during the Quadrature in all the monitoring points. In the results of Syzygy for SLR of 50 and 100 years, low negative variations (0.28–3.83%) were observed in points A and C and positive variations (3.39–8.25%) in point B. In this sense, despite of the slight decrement in the velocity currents, we might state that the overall effect of SLR is positive over currents, showing a positive trend as the years go by.

Considering that SLR produces increments in velocity currents, it also may provoke changes in the morphodynamics of Buenaventura Bay. These changes may not only affect the intertidal areas, but also the hydrodynamic field because of the modification of the bottom shape and the estuaries. Then, the identified effects of SLR on the velocity currents shown in this study are limited by the assumptions that the shape of the bay will not change significantly. In addition, this study considered constant temperature and salinity in terms of vertical distribution and time, as well as the discharge of the Dagua River, because the deepening and SLR cases were analyzed in a long-term time horizon that allows the assumption that short-term variations will not be significant. Although the assumption of this study was handled with the proper calibration and validation of water levels and currents, it is recommended in future studies to evaluate the effect of short-term variations of thermohaline parameters and river discharge.

Considering that point A showed the highest mean velocity currents for all the numerical cases, it was hourly analyzed for 2021 (Figure 7). The location evidenced that velocity currents followed the tidal regime where the spring and ebb tides reported velocities about 1.1 m/s and 1.7 m/s during the entire year; as expected, the currents of ebb tides tend to be higher than spring tides considering that during ebb tides the bay is drained.

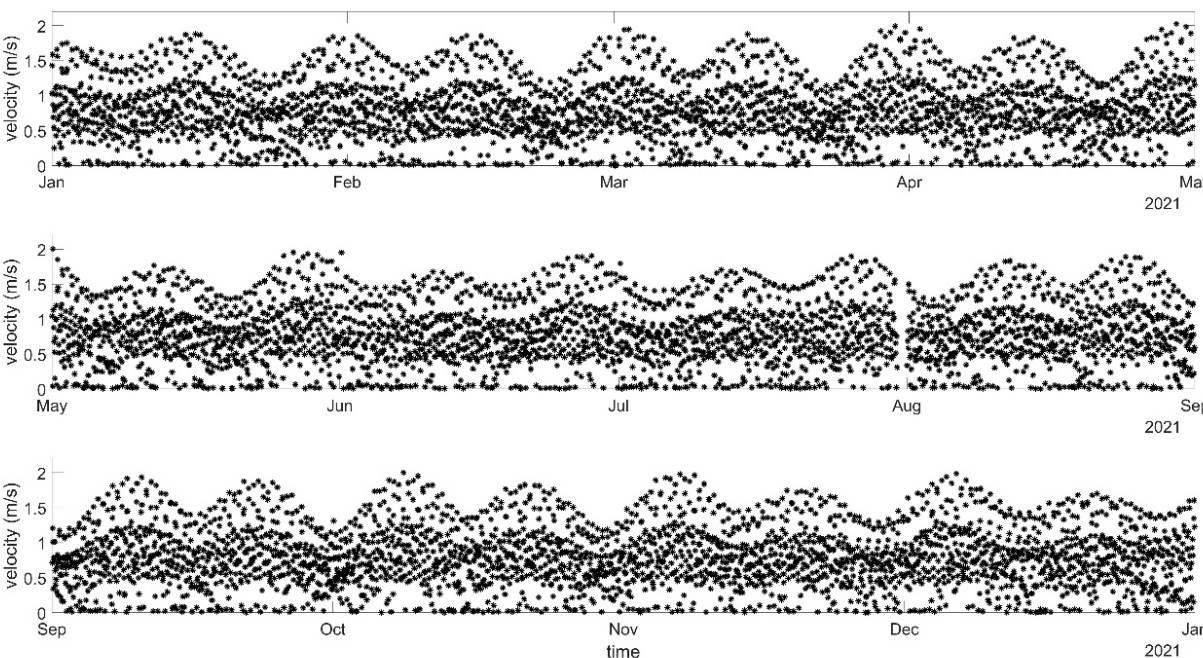

**Figure 7.** Depth average modeled velocity currents of point A during 2021 in Buenaventura Bay.

The hydrodynamic results of our study showed similar patterns reported in the literature, related to the effect of sea level rise and changes in bathymetry on semidiurnal meso-tidal range estuaries. The study of Chen and Liu [34] analyzed several tidal stations with semidiurnal cycles in the Taiwan Strait, where the Taichung Harbor station has a tidal range near to 4 m, similar to our study. That research reported that sea level rise increased the current velocities during all the cases and agreed with the effect of sea level rise seen in our work. In this sense, the sea level rise reduces the bottom friction of the estuaries, as Khojasteh et al. [32] commented, easing the increment of surface current velocities in Buenaventura Bay.

Alvarez et al. [52] studied the influence of dredging on future tidal Stream Energy farms in a partially mixed estuary of Spain known as Ría de Riabadeo. The study area has a semidiurnal tidal regime (F = 0.080) and maximum tidal range up to 4.6 m, similar to Buenaventura Bay in Colombia. Their study reported depth-averaged tidal flow velocities in the range of 0 to 1.25 m/s and pointed out that dredging activities on the estuary affected the currents and reduced by more than 10% the rated power of the stream tidal turbine. In line with this, the access channel deepening of our study reduced by about 14.28% the velocity currents (Figure 6), similar to the findings of Alvarez et al. [52].

The effect of access channel deepening in this research was observed in the decrement of the current velocities in Buenaventura Bay, mainly in the areas near to the channel. The reduction of magnitude in the tidal currents may be explained in two forms. In the first, the control volume of the access channel increase provokes the vertical surface control area rises, and current velocities reduce. The second explanation considers the bay as a hydraulic channel: when offshore ocean currents enter the bay through a downward step and pass over a deepened channel, the flow velocities reduce because of the increment of water depth of the channel; in terms of energy balance, the increment of potential energy (water depth) reduces the kinetic energy of the channel. The aforementioned explanations must be analyzed in further studies through hydrodynamic modeling cases varying the access channel depth and other hydraulic properties of the bay.

Other studies have evaluated electricity generation through hybrid systems, considering the use of tidal turbines. Zhou et al. [53] performed a techno-economic assessment of a theoretical hybrid renewable energy system in Hong Kong City, compounded by tidal

turbines and floating PV generators. They calculated tidal power with monthly current velocities between 0.76 and 0.84 m/s and reported that integrating floating PV and tidal stream turbines is a feasible alternative for producing electricity. Abdullad et al. [54] also recommended hybrid systems such as solar PV panels integrated with tidal stream turbines to attend efficiently to the energy demand. Those hybrid systems worked with marine currents with similar velocity ranges to those of Buenaventura Bay; hence, the use of hybrid systems integrated by third generation stream turbines could increase the feasibility of tidal power plants in Colombia.

## 4. Conclusions

This research aimed to increase the understanding of hydrodynamics in Buenaventura Bay and the effect of sea level rise and bathymetry changes due to access channel deepening. The calibrated and validated numerical results of water levels and velocities evidenced that the study area is a hyper-synchronic estuary, where the convergence effects are higher with respect to the frictional effects generating tidal amplification mainly in the upper reaches. Therefore, this work explained why previous studies showed differences between the modeled and measured amplitudes during high and low tides of 1 m and 0.2 m, respectively. To handle the abovementioned, it was necessary to add extra tidal harmonics for considering the amplitude amplification and avoiding water level differences.

The 24 monitoring points within Buenaventura Bay have shown mean velocities between 0.25 and 1 m/s in normal conditions, which increases the possibility of implementing tidal stream turbines to operate in the tidal range of 3–4 m with semidiurnal cycles. The sea level rise increases the current velocity in the bay, and the access channel deepening had the opposite effect; consequently, these results may be considered valuable for the planning of future tidal power plants, as well as the enhancement of port activities. This study is the first of its kind in depicting tidal currents with a fully calibrated hydrodynamic model; hence, our results open the door for the better understanding of other physical and biochemical processes such as sedimentation–erosion generated by the rivers' discharges in the study area and changes in the water quality due to domestic–industrial pollutant discharges.

We recommend for the planning of future tidal plants in sheltered bays to verify the presence of tidal amplifications, because the correlation coefficient derived from the comparison of modeled–measured data is not sufficient to properly simulate the natural hydrodynamics of hyper-synchronic estuaries such as Buenaventura Bay. If the tidal amplification is not corrected, the numerical models will produce the under-over estimation of velocity currents affecting the technical and economic performance of the tidal stream turbines. Finally, the reported values of current velocities and the effect of changes in the bathymetry and sea level rise of this study make Buenaventura Bay a candidate for the implementation of third generation tidal stream turbines and motivate future research for developing new turbines capable of operating in low velocity conditions.

**Author Contributions:** Conceptualization, J.G.R.-B.; methodology, J.G.R.-B.; software, J.G.R.-B.; validation, J.G.R.-B.; formal analysis, J.G.R.-B.; investigation, J.G.R.-B.; resources, J.G.R.-B.; writing—original draft preparation, J.G.R.-B.; writing—review and editing, J.G.R.-B., J.L.G.V. and D.M.P.-V.; supervision, J.G.R.-B.; project administration, J.G.R.-B. All authors have read and agreed to the published version of the manuscript.

**Funding:** This research received no external funding.

**Data Availability Statement:** The measured oceanographic data provided by DIMAR is available in CECOLDO database https://cecoldo.dimar.mil.co/web/ (accessed on 8 March 2023).

**Acknowledgments:** The authors offer thanks to the Maritime General Directorate of Colombia (DIMAR, in Spanish) for providing the measured hydrographic and oceanographic data.

**Conflicts of Interest:** The authors declare no conflict of interest.

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
