# Peer review of "Effect of Sea Level Rise and Access Channel Deepening on Future Tidal Power Plants in Buenaventura Colombia"

_infrastructures, doi:10.3390/infrastructures8030051_

Round 1

Reviewer 1 Report

This paper discussed the effect of sea level rise and channel deepening on future tidal power plants in Buenaventura Colombia. The topic of this paper fits within the stated scope of this journal and I think readers may be interested in this study. Several aspects could be further improved in order to having it published in this journal. Therefore, my recommendation is major revision. The main questions I encountered when reading the manuscript are as follows:

Major technical concerns:

1、The setting of open boundary conditions is very important, but the authors do not describe their setting in detail. How many and what tidal constituents are used? Why not use widely-used tidal models such as TPXO9 or FES2014 as open boundary conditions? I do not understand why three additional tides are needed to add for the open boundary. As displayed in Table 3, harmonic with periods of 12 and 12.42 hours are S2 and M2 tides which are the most vital constituents. Why the authors need to find their periods via trial-error? The periods of S2 and M2 are common knowledge!

2、There are numerous problems in Table 2. Tidal periods shown in Table 2 are not correct. For example, the period of S2 tide is exact 12 hours not 11.9986 hours. What are the meanings of A, B, C? The authors do not explain. Also, T1 and Sa tides have same tidal frequencies. I recommend the authors use tidal harmonic analysis software such as S_TIDE or T_TIDE to derive tidal harmonic constants rather than Fourier analysis. The authors can use s_tide_m8 function in S_TIDE (https://www.researchgate.net/project/A-non-stationary-tidal-analysis-toolbox-S-TIDE) to perform harmonic analysis and examples are displayed in s_demo.m.

3、I think the authors should draw co-tidal charts for different cases, which can clearly indicate the potential effects of sea level rise and channel deepening on tidal dynamics. The authors can refer s_draw_tidalchart function in S_TIDE to know how to draw co-tidal charts.

Major editorial concerns:

1、Title:I do not understand the meaning of access in the title, please revise

2、Abstract is not well-written. The authors introduce too much about background in the abstract. In fact, I think only one or two sentences are needed on background. The focus should be on introducing your work and new findings.

3、Introduction is not well-written. Some paragraphs are abnormally long, please revise. Your introduction does not clearly indicate the motivation of this study. Does the study area undergo a rapid sea level rise and severe channel deepening? The authors need to provide more information.

Minor concerns:

Language can be further improved.

L22: change the to tidal

L30:change affect to affects

L31:change put to puts

L36: change when to since

L37:delete current because most ocean currents are driven by tides

L38:delete the

L45-47:please revise this sentence

L73:change exist to exists

L76:change research to researches

L82:change simulate to explore

L84: change have to has

L95:change  through to via

L100: delete the before sea level rise

L103: delete changes in

L136: The appears twice

L143: Figure 1 is not clear

L152: delete the before Figure 1

L370: add should before verify

L372: change ‘simulate properly’ to properly simulate

L373: add it before is

Author Response

Dear reviewer, we thank your valuable and timing comments, because they help us to improve the manuscript. Please see in the attached file the reply to your concerns.

Reviewer 2 Report

Brief summary:

This work examines the potential for tidal energy generation in Buenaventura Bay in Columbia. The authors use a numerical model, validated with water level and velocity data, to examine the changes to the tides from a channel deepening of 5 m (dredging for navigation) or a sea level rise of 0.44 m, the projected increase for the region in 200 years. After looking at changes to three selected sites when compared to the 2021 tidal variability, increases in maximum tidal velocity were seen for the sea level rise scenario, and decreases were seen for the channel deepening scenario.

The biggest strength of the manuscript is the tuning of the model to the local data, improving model-data fit, and the interest of the topic, which is an increasingly important one as we try to limit fossil fuels and respond to changes in climate. The weaknesses include the inability to clearly demonstrate that the projected changes would impact the ability to generate tidal power one way or the other, overall the lack of connection between the literature cited and the specific results in the paper, and data figures that sometimes obscure rather than clarify the points being made. In order for this to be ready for publication, statistical analysis of the final conclusions need to be included in a substantial way, to show that the changes shown are significant and clarify the magnitude of the changes, and the text needs to more clearly discuss the implications as they connect to the project goals.

Detailed comments:

Abstract: There is a reference to effects of the relevant processes on time scales of days or decades, but the research investigates a 200-year timescale. These should better align.

Line 44-47: Is there a need for this history lesson, or can the audience be assumed familiar with the development of tidal energy? In this paragraph overall it would benefit from more focus on the sea level ranges and velocity magnitudes that are sufficient for generating tidal energy, both with projects in place and those planned for the near future.

Lines 47-57: Similarly, this just seems like a list of locations. A summary, or even a table, might more effectively make this point. The concluding sentences of this paragraph feel more appropriately summarized.

Lines 76-86: What is the motivation of the listing of these models and calculations? There should be a clear connection to the research being proposed.

Line 111: In concerns over sea level rise, what is the rate of increase of sea level, and is it enough to impact the calculations of tidal power generating capability? A small change in depth will not be likely to have huge impacts on tidal resonance, if we are considering centimeter changes over decadal time scales.

Line 125: The study goals are worthy ones, to look at the potential for renewable energy generation through tidal power along the coast of Columbia.

Line 139: Wikipedia is not a valid scientific source. Information may be changed over time.

Figure 1: The A, B, and C labels on the map are hard to read, consider making the text larger. I would suggest moving tidal station and ADCP measurement coordinates out of the figure caption, and into a table or other location.

Line 149: Delft3D mis-spelled

Line 136: What are the deepest depths within the Bay/study area? This is also relevant for assessing the utility of 10 sigma layers, based on the total water depth.

Line 160: Given that this is not an open-ocean region, how consistent are these properties? Are the data by month/week/etc, what is the time resolution of the WOCE database for this region? Were any density data collected for validation purposes?

Line 164: Should there be a database citation for the DIMAR data?

Line 173: I agree that 2.2 mm/year is a reasonable choice. How long would a tidal energy installation be expected to last? Is it more or less than 200 years? If equipment would be replaced/upgraded prior to that time, then a shorter horizon needs to be used, which would reduce the magnitude of anticipated changes in sea level.

Line 188: In its current format, I’m not sure that Figure 2 adds much to the understanding of the data. It looks like there is a table below with results of a proper harmonic analysis (examining tidal frequencies to focus on those with highest amplitudes), the periodogram is not easy to interpret as presented. If kept, consider showing a noise ceiling or reducing the period shown, and don’t use the MATLAB tooltip to show numbers, but instead add them as text with relevant information to focus the reader’s attention.

Line 203: The abbreviation is ITCZ

Table 2: I suggest adding a column showing signal to noise ratio, to highlight which constituents have a significant signal. This would be particularly useful for the longer period signals identified. Please clarify what A, B and C represent in column headers: since this is water level data, I was expecting to see amplitude, but some values are negative, so I don’t know what is meant by these coefficients and it is not explained in the text or table caption. Consider sorting by signal to noise ratio, or highlighting the significant constituents.

Line 199: As noted above, I would like to see more justification that these “new” signals are significant, and perhaps more investigation of what meaning they may have if not associated with a known tidal frequency.

Line 215: I have concerns if this model is off by up to 1 m amplitude in tidal elevation during spring tides, if the model is being used to make projections focused mainly on tidal amplitude in terms of potential for energy generation. I understand that corrections were made to the model that more closely match it to the data in hindcasting, but if the system is changing over a much longer timescale, can you be confident this will not affect the results?

Figure 3: you need to change the symbols in (b), right now it is just a giant blob and I can’t really tell if the model and measurements agree or not. Either make the symbols much smaller, or omit them altogether and just use a connecting line, as the resolution is quite high compared to the length of measurement and can’t really be resolved by eye either way. You can leave the symbols for the inset (zoom-in), where they can actually be resolved. Perhaps it would be helpful to also show the comparison prior to the adjustments to help show that the fit is improved?

Line 252: Since it has not been stated what the total water depth is, is 10 m depth a good match for the first sigma layer?

Line 254: Do you mean to say p < 0.05, as stated in the figure, rather than p = 0.05?

Figure 4: Panel (b) is difficult to read, the resolution is not very good and it’s very compressed. Consider a different formatting choice. What is meant by records on the x-axis? Time would be more intuitive.

Figure 5: There seem to be two Figure 5’s, I’m not sure if the same figure (a) and (b) are included twice? Figure 5 is bolded in the text making it look like a figure caption, but I assume that is just a regular paragraph starting on line 272.

Line 299: The trends of increased and reduced velocities appear valid, but are they statistically significant? What is the magnitude of the change, and is that meaningful not only statistically but in terms of impacts to the potential for energy generation? Sites B and C seem unlikely to be strong enough flow regardless. Would it make more sense to select three locations with relatively high velocities rather than a high, medium and low velocity site?

Line 215: Again, during syzygy, are any of these changes significant statistically or in terms of potential to impact energy generation? And how sensitive is the tidal damping to the specific amount of channel deepening? Also, with the deepening, is the channel only dredged, or is the model increasing depth throughout the domain of the Bay? This would impact the hydrodynamics.

Line 329: This feels like a sudden transition from results to discussion, putting your findings into context with prior research. Consider adding a separate section or sub-section when transitioning from direct reporting of results to interpretation.

Lines 336-346: The statements here follow the results reported previously, that overall velocities increased with the sea level rise scenario and decreased with the channel deepening, although how significant/meaningful those changes are has not yet been addressed. It seems a bit contradictory that the increase in sea level is cited as a decrease in friction, while the channel deepening might have the same effect, but instead is thought of as a control volume increase – why would this not also factor into the sea level rise, then?

Line 373: Are there any known considerations of generating tidal energy in this area specifically?

Author Response

(The authors gave the same response as above.)

Reviewer 3 Report

The manuscript, which is overall easy-to-follow, aims to increase the understanding of hydrodynamics of Buenaventura Bay (Colombia) and the effect of sea level rise and bathymetry changes due to the access channel deepening. A 3D hydrodynamic model was first calibrated against data recorded by three monitoring points, and in particular three harmonics were added to correct the modelled water level amplitudes. Then the model was used to simulate the velocities in the Bay in three modelling cases, namely normal, sea level rise and deepening conditions. It was found that the sea level rise increases the currents velocity, whilst the access channel deepening has the opposite effect, as expected. The work is an interesting reading, and the Reviewer thinks that only a minor revision is needed before possible publication. The detailed comments are presented below:

1)      Line 80. The acronym “CFD” should be spelled out.

2)      Line 116. The concepts of Sicigy and Quadrature should be briefly explained.

3)      Line 151. Please explain the meaning of sigma layers.

4)      Figure 3 is split among two pages. It is suggested to create figures embedding all subplots and letters a), b), c) etc, so that they will not be split when loaded in Word.

5)      It seems that Figure 5 is introduced twice, and for its second occurrence the letters a) and b) are separated from the subplots. Please amend. Also, units in the colorbar legend should be added.

6)      Since the simulation results of Section 3.2 are referred to the lowest Quadrature and the highest Sicigy, whose ebb tides are shown in Figure 5, it is suggested to move the latter figure to Section 3.2.

7)      A text proofreading is needed to improve the quality of the English language and the readability of several sentences, as well as to fix typos throughout the manuscript (e.g., “The The domain area…”, line 136; “de capability”, line 250).

Author Response

(The authors gave the same response as above.)

Round 2

Reviewer 1 Report

 I am not satisfied with the revisions. Tidal periods shown in Table 2 are not correct and the authors do not change them. Tidal periods are well-known and the authors can find them in any textbook of tides or widely-used tidal harmonic analysis toolbox (e.g. T_TIDE, U_TIDE, S_TIDE).  I want to emphasize that tidal periods are constants and not changed by friction or other factors. Due to uncorrect tidal periods of constituents in the experiments, thus, the authors have to add three additional tides to correct errors. Why not set correct tidal periods in your experiments?

If the authors do not change their tidal periods in the next revision, I will reject this paper.

Author Response

Dear reviewer, we thank the time that you have provided for conducting the review of our manuscript. Please see in the attached file the response to your review.

Reviewer 2 Report

The revisions improve the paper substantially. Below are comments on areas I still feel need to be addressed, particularly making sure that the statistical significance as well as the practical significance of the findings are clear to the reader.

Abstract: much clearer, however would either of your scenarios lead to Buenatentura Bay no longer being a candidate for tidal power generation? That seems like the main goal, and I am still not clear whether these impacts are large enough to either slow velocities enough to the point that currents are too weak for tidal power generation, or speed them up enough to make it a better candidate for this type of renewable energy. Also, if the fastest velocities are in the channel, which is maintained for shipping, would that be an issue for the placement of a tidal generator?

Importance of sea level rise and time horizon: I understand that 200 years was selected as a timeframe that would produce appreciable changes, and that there could be significant societal impacts for a number of reasons under this scenario. However, I am still not convinced that this is a helpful timeframe for this problem as posed, given (1) that technology will likely advance over the next 200 years, meaning we may be able to generate tidal energy with lower velocities, and (2) uncertainty of the linearity of the processes increases with the time horizon. Ideally, you would select several scenarios to show how the changes scale with changes in sea level, to better demonstrate the point where significant changes emerge.

On the stratification properties: My concern is that in an estuary, salinity and temperature profiles vary on tidal cycles, so no database will cover all scenarios, unlike the open ocean where a seasonal profile will remain fairly representative of conditions over time. I suggest adding a comment on how the variability of temperature and salinity in the system may affect your model results.

Tidal harmonics: I appreciate the clarified labeling in the table showing the tidal harmonics. You did not however address my question about the signal-to-noise ratio. I see from the rearranged table that the M2, as expected, has the highest amplitude signal, and the T1 signal you identified is the same order of magnitude as the other diurnal and semidiurnal constituents. However, as the signals move towards lower frequency, the resolution also decreases due to the length of the data signal, and so it is more difficult for a result to be significant. If you report signal-to-noise ratio, which is an output of common tidal analysis software, the reader can better determine if these signals rise above the background noise. I would find it surprising if the T1 signal is significant, which would depend on both the record length and the noise levels of the data. Given that these “new” signals are a minor point in your study, if you cannot show that they are statistically significant, I would suggest de-emphasizing that result and possibly cleaning up your table to only show significant tidal constituents. I understand that the constituents are listed in order of tidal period, and that you calculated the F-factor, but again these are separate considerations from my point of concern.

Table 4 is very helpful. It seems quite meaningful that the deepest site (which would likely be the best candidate for tidal energy) was least impacted by the channel deepening, and in fact showed a slight increase in velocity during both spring and neap tides.

Author Response

(The authors gave the same response as above.)

Round 3

Reviewer 1 Report

This paper can be accepted.